# Hybrid AI Approach for Counterfactual Prediction over Knowledge Graphs for Personal Healthcare

Hao Huang
hao.huang@tib.eu
TIB - Leibniz Information Centre for
Science and Technology
Hannover, Lower Saxony, Germany

Emetis Niazmand
emetis.niazmand@tib.eu
TIB - Leibniz Information Centre for
Science and Technology
Hannover, Lower Saxony, Germany

Maria-Esther Vidal
maria.vidal@tib.eu
L3S Research Center
Hannover, Lower Saxony, Germany

## ABSTRACT

Artificial intelligence (AI) has become an invaluable tool in healthcare for disease prediction and diagnosis. Despite their predictive accuracy, AI models may ignore causal relationships between patient characteristics (demographic or clinical). As a result, although AI models capture associative patterns, ignoring the causal relationships of predicted characteristics limits their ability to perform counterfactual reasoning about the predicted characteristics of a patient when dependent characteristics change. This limitation of AI models can affect the understanding of predicted outcomes. We have proposed hybrid AI methods that combine symbolic reasoning over knowledge graphs (KGs), large language models (LLMs), and causal reasoning techniques to infer causal relationships between patients' properties. As a result, a causal model is learned, enabling counterfactual prediction to support clinical decisions. We apply these AI methods to predict the counterfactuals of biomarker results in non-small cell lung cancer (NSCLC) patients under hypothetical treatments with different smoking habits. We have created synthetic datasets based on clinical records of NSCLC patients to evaluate the performance of the proposed methods. The observed results suggest that our methods are competitive with baseline methods in causal relationship discovery and counterfactual prediction.

## CCS CONCEPTS

• **Applied computing** → **Health care information systems**; • **Computing methodologies** → **Artificial intelligence**.

## KEYWORDS

Knowledge Graphs, Counterfactual Reasoning, Healthcare

**ACM Reference Format:**
Hao Huang, Emetis Niazmand, and Maria-Esther Vidal. 2024. Hybrid AI Approach for Counterfactual Prediction over Knowledge Graphs for Personal Healthcare. In *Proceedings of AI and Data Science for Healthcare: Bridging Data-centric AI and People-Centric Healthcare (AIDSH 2024)*. ACM, New York, NY, USA, 6 pages. https://doi.org/XXXXXXX.XXXXXXX

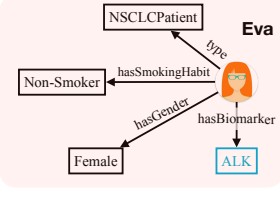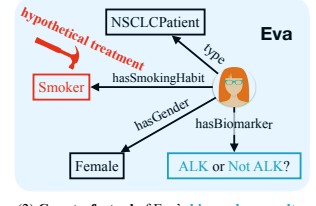

(1) **Factual** of Eva's biomarker results

(2) **Counterfactual** of Eva's biomarker results what if she were smoker

**Figure 1: Motivating Example.**

## 1 INTRODUCTION

Explainable Artificial Intelligence (XAI) has emerged as a paradigm to facilitate the interpretability and transparency of traditional machine learning (ML) models [4, 17]. In healthcare, XAI facilitates clinical decisions with AI-driven insights [2]. Recent studies investigate the great possibilities of integrating AI in diagnostic and prognostic processes [5, 6, 10], highlighting the potential of AI to transform medical practices. A key aspect of making AI insights actionable in healthcare is the understanding of causal relationships and counterfactual scenarios [24]. Causal inference and counterfactual prediction go beyond simple correlations, providing deep insights into the "*why*" and "*what if*" behind AI decisions, essential for clinical applications [22]. Despite advances, these techniques assume the availability of causal knowledge, such as relationships between variables, representing patient characteristics (e.g. age, sex, and biomarkers). Causal associations can be obtained from domain experts, learned from data [29], or extracted from knowledge bases such as ontologies and knowledge graphs (KGs) [23]. KGs [13] integrate data and metadata (specified in ontologies), which encode the meaning of properties, such as their domain and range. Consequently, both ontologies and KGs provide contextual information crucial for discovering causal relationships and facilitating counterfactual reasoning in the medical domain.

**Motivating Example**. Consider a non-small cell lung cancer (NSCLC) patient, Eva, who is a female non-smoker and tested positive for the anaplastic lymphoma kinase (ALK) biomarker. Figure 1-(1) partially visualizes Eva's characteristics, including her gender, smoking habits, and biomarker results. The oncology researchers aim to investigate how smoking habits affect biomarker results in patients like Eva. Specifically, they want to examine the hypothetical outcome of Eva's biomarker results if she were a smoker, shown in Figure 1-(2). Current AI techniques in healthcare [5, 6, 10] rely on predictive models, such as the random forest [5] and deep learning models [10], to make accurate predictions. However, *"Can we simply modify Eva's smoking habits in the dataset and use a predictive model to predict her counterfactual biomarker outcome?"* The answer

is *Yes*, if the model accurately captures causal mechanisms, i.e., causal dependencies between patient characteristics (or properties). However, as Pearl points out [22], *"Data does not understand cause and effect; people do."* Predictive models trained on observational data typically lack the ability to identify causal relationships. Pearl [21] introduces the twin networks, where one network models the causal mechanism in the actual world (factual) and the other models the mechanism under hypothetical scenarios (counterfactual), illustrating that a single predictive model trained only on factual data may not able to represent the different causal mechanisms.

**Problem Statement**. Given a patient (entity) $e$ in a KG and treatment and outcome properties $p_T$ and $p_Y$ ($p_T$ is assumed to have a direct impact on $p_Y$). This work aims to predict counterfactuals in $p_Y$ for patient $e$ when $p_T$ of $e$ is altered, while keeping all other properties of $e$ constant. Our objective is to learn a causal model that can accurately predict the counterfactual outcomes on $e$ given contextual information and a hypothetical treatment of $e$. As shown in our motivating example, this involves determining the outcomes of NSCLC patients under hypothetical changes in smoking habits.

**Proposed Solution**. We propose hybrid AI methods that combine: (1) symbolic reasoning on the metadata modeled in the ontologies of KGs; (2) numerical approach implemented in LLMs and ML methods; and (3) causal discovery and reasoning. As a proof of concept, we implement the hybrid method over a medical KG that integrates clinical data from lung cancer patients for counterfactual reasoning. This KG was built following the methods proposed by Vidal et al. [27] and Aisopos et al. [1]. Due to privacy restrictions, the results presented in this paper were derived from synthetic KGs created from the original KG reported in [1, 27]. The metadata captured in the medical KG is used to improve the accuracy of causal discovery and the performance of counterfactual prediction.

**Evaluation**. We evaluate the effectiveness of our method in a use case of predicting the counterfactual biomarker (outcome) of NSCLC patients if the patient's smoking habits (treatment) were changed. In each dataset, the factual and counterfactual biomarker outcomes of patients are generated using an Additive Noise Model (ANM) [14] learned from the original data and based on expert-designed causal knowledge, including all causal relationships between the characteristics of NSCLC patients. Our empirical results show that our method outperforms other methods in discovering causal relationships and is competitive in counterfactual reasoning.

**Contributions**. The contributions of this work are: (1) The hybrid AI methods for causal relationship discovery over KGs. (2) Empirical evaluation of the proposed method on synthetic KGs created from clinical data of patients with NSCLC. This paper is structured as follows: Section 2 presents the foundational concepts for the proposed approach. Section 3 defines the problem statement and introduces the proposed hybrid AI method. Section 4 details the experimental settings, reports the used metrics, and discusses the results. Section 5 analyzes the state-of-the-art. Finally, section 6 summarizes our findings and explores future directions.

## 2 PRELIMINARIES

**Knowledge Graph [13]**. A KG is a *directed edge-labeled graph* $KG = (V, L, E)$, where $V, L$ are respectively a set of entities and property labels, and $E \subseteq V \times L \times V$ is a set of triples. A $KG$ metadata is part of

$KG$ and corresponds to triples $E^* = \{(s, p, o) \in E | p \in L^*\}$ expressing the meaning of data, where $L^* \subseteq L$ is a set of properties used to annotate entities, classes, and properties.

**Ego Network [11]**. Given a knowledge graph $KG = (V, L, E)$, an entity $e \in V$, the *ego network* of $e$ in $KG$ based on a set of properties $P \subseteq L$ is defined as $\phi(e, P) = \{(e, p, o) | p \in P \wedge (e, p, o) \in E\}$.

**Causal Concepts**. In a causal analysis [18], each unit $e$ refers to an entity subjected to an intervention $do(T=t)$, where $T$ is a binary variable indicating treatment status, which is either the active treatment $T=1$ intended for the impact analysis, or the control treatment $T=0$ as reference for $T=1$. The treatment effect of $T=1$ versus $T=0$ on unit $e$ is evaluated by comparing the corresponding potential outcomes $Y(1)$ and $Y(0)$ of $e$. The factual outcome under the actual treatment $t$ is $Y^F = Y(t)$, while the counterfactual outcome under the hypothetical treatment 1-$t$ is $Y^{CF} = Y(1-t)$.

**Causal Graph**. A *causal graph* [19] is a *directed acyclic graph* (DAG) $G = (X, E')$, where $X$ is a set of variables and $E' \subseteq X \times X$ are the causal relationships between variables, where $(X_i, X_j) \in E'$ if the variable $X_i$ causes $X_j$. Given a DAG $G$, we denote $edges(G) = E'$.

**Causal Bayesian Network [21]**. Given a *causal graph* $G = (X, E')$, let $P(X)$ be the joint distribution over variables $X$ following the factorization according to $G$, specifically, $P(X) = \prod_{X_i \in X} P(X_i | Pa_G(X_i))$ where $Pa_G(X_i) = \{X_j | (X_j, X_i) \in E'\}$. Let $P_t(X) = P(X | do(T=t))$ be the distribution under intervention $do(T=t)$ ($T \subseteq X$). The tuple $(G, P_t)$ is a Causal Bayesian Network (CBN) [21] iff: (1) the distribution $P_t(X)$ over $X$ follows the factorization according to $G$; (2) $P_t(T_i = t_i) = 1$ for all $T_i \in T$, whenever $T_i = t_i$ consists with the intervention $do(T=t)$; and (3) $P_t(X_i | Pa_G(X_i)) = P(X_i | Pa_G(X_i))$ for all $X_i \in X \setminus \{T\}$. With conditions (1-3), the interventional distribution $P_t(X)$ under any intervention $do(T=t)$ is computed with a truncated factorization [21]: $P_t(X) = \prod_{X_i : X_i \in X \setminus \{T\}} P(X_i | Pa_G(X_i))$.

## 3 OUR APPROACH

Let $KG = (V, L, E)$ be a KG, $C$ be a class s.t. $(C, type, Class) \in E$, and $P_C = \{p | (p, domain, C) \in E\}$ be all properties of $C$. We introduce some new definitions necessary for our approach.

**Dataset over KG**. The dataset of $C$ based on $P_C$ and $KG$ is defined as $D_{KG}(C, P_C) = \{\{(p, o) | (e, p, o) \in \phi(e, P_C)\} | (e, type, C) \in E\}$.

**Causal Concepts over KG**. Given two properties $p_T, p_Y \in L$ representing treatment and outcome, s.t. $(\forall p \in \{p_T, p_Y\})[(p, domain, C) \in E]$. The set of contextual properties of $C$ based on $p_T$ and $p_Y$ is defined as $P_C(p_T, p_Y) = P_C \setminus \{p_T, p_Y\}$. Each entity $e \in V$ s.t. $(e, type, C) \in E \wedge (e, p_T, t) \in E \wedge (e, p_Y, y) \in E$ is defined as a unit. The triples $(e, p_T, t), (e, p_Y, y) \in E$ are the treatment and outcome of $e$, and the *ego networkk* $\phi(e, P_C(p_T, p_Y))$ is the context of $e$.

**Causal Model over KGs**. Given the treatment and outcome properties $p_T$ and $p_Y$, and the set of contextual properties $P_C(p_T, p_Y)$. Let $E_T$ be a set $V \times \{p_T\} \times V$, and $E_Y$ be a set $V \times \{p_Y\} \times V$. A *causal model* of $p_Y$ is a function $\vartheta : E_T \times \mathcal{P}(E) \rightarrow E_Y$, s.t. $\vartheta((s, p_T, t), E'') = (s, p_Y, y)$, where $\mathcal{P}(\cdot)$ denotes the powerset of a set, $\{s, t, y\} \subseteq V$, and $E'' = \{(s, p, o) | (s, p, o) \in E \wedge p \in P_C(p_T, p_Y)\}$.

**Problem of Counterfactual Prediction over KGs**. Given a medical KG $KG = (V, L, E)$, a target class $C$, all properties of $C$, $P_C$, including treatment and outcome properties $p_T, p_Y$, and contextual properties $P_C(p_T, p_Y)$. Let $V_T$ be a set $\{o | (s, p_T, o) \in E\}$. The problem of counterfactual prediction is to find an optimal *causal model*

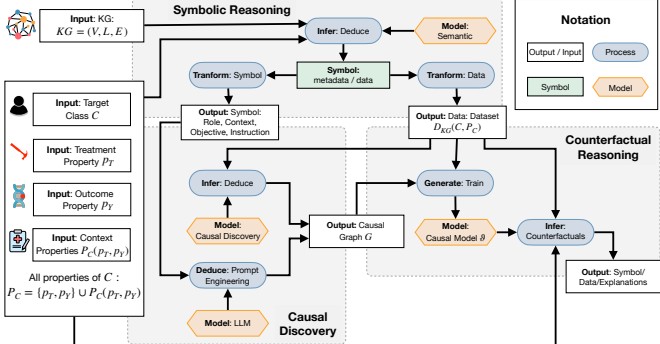

**Figure 2: HealthCareAI Design Pattern based on patterns proposed by Bekkum et al. [26].**

$\vartheta^*$ for $p_Y$ in the space of all possible *causal models* $\Theta$, s.t. for each patient $e$, the *causal model* $\vartheta^*$ predicts the counterfactual outcome of $e$ with the highest probability among all possible outcomes, given its context $\phi(e, P_C(p_T, p_Y))$ and a hypothetical treatment $(e, p_T, t')$ s.t. $t' \in V_T \wedge (e, p_T, t') \notin E$.

**Our Solution**. To solve the problem, we propose hybrid methods exploiting both data (all properties of patients, $P_C$) and metadata (encoding the meaning of properties and classes) in $KG$. Our approaches establish causal relationships among properties in $P_C$ using data-driven and metadata-driven (e.g., LLM) causal discovery models, laying the foundation for a causal model $\vartheta$ to capture the causal mechanism, which generates the outcome property $p_Y$ of patients. The model $\vartheta$ is trained to maximize the likelihood of the observed data (in $KG$), enabling it to learn a causal mechanism that is optimized to predict the most probable counterfactual outcomes of each patient under a hypothetical treatment.

**A Design Pattern for HealthCareAI**

The design pattern of our proposal is illustrated in Figure 2, following the design principles by Bekkhum et al. [26], where the white rectangles represent outputs and inputs; the green rectangle represents symbol; the blue ovals represent processes; and the orange hexagons represent models. It comprises three components that tackle the tasks of causal relationship discovery and *causal model* learning towards solving the problem of counterfactual prediction.

**(1) Symbolic Reasoning**. This component takes as input a medical KG $KG = (V, L, E)$, a target class $C$ (e.g. NSCLC Patient), and all properties of $C$, i.e., $P_C$. It employs a semantic model, which defines the meaning of symbols in KG, to perform reasoning and query processing over KG. This process deduces data and metadata for the counterfactual prediction. Specifically, it queries the properties $P_C$ (data) of all patients (units), and transforms the data into a dataset $D_{KG}(C, P_C)$. It also extracts the metadata that describes the specific domain of the medical knowledge graph $KG$ (e.g., lung cancer), the meaning of the target class $C$, and the properties in $P_C$. The metadata is used to populate an LLM prompt [7] for querying causal relationships between properties in $P_C$. Although generic, we consider basic metadata about properties (e.g., domain, range, and notation) and classes. However, as shown by Hung and Vidal [15], other reasoning methods can be used to improve causal reasoning.

**(2) Causal Discovery**. This component aims to infer the causal graph or DAG that encodes the causal relationships between properties in $P_C$. It takes as input the prompt sections for querying causal

relationships and the dataset $D_{KG}(C, P_C)$ derived from $KG$; and outputs a causal graph $G$. To do this, it uses a data-driven model **causal discovery** and a metadata-driven model **LLM**. The data-driven method can be any traditional causal discovery method, such as **PC** [25] and **FCI** [25], or **GES** [8], which uses the dataset $D_{KG}(C, P_C)$ and produces a causal graph $G_1 = (P_C, E_1)$. The metadata-driven modal can be any LLM that takes the LLM prompt as input and outputs a causal model $G_2 = (P_C, E_2)$ represented by a set of causal relationships. We designed four sections for the LLM prompt, where the role section uses the metadata describing the domain information of $KG$ for defining the persona or function the LLM should adopt; the context section uses the metadata describing the meaning of the target class $C$ and properties in $P_C$. For example, the domain (rdfs:domain) and range (rdfs:range) of properties, the human-readable label (rdfs:label) and annotation (rdfs:comment); the objective section specifies the task of identifying causal relationships between properties in $P_C$; the instruction section formats the output of the causal relationships. The final output of this component is a causal graph $G = (P_C, E')$ s.t. $E' = E_2 \cup \{(c, e) \in E_1 | (P_C, E_2 \cup \{(c, e)\})\}$ is a DAG}. In other words, it includes all causal relationships in $E_2$ (by **LLM**) and those in $E_1$ that do not introduce any directed circle in $G$.

**(3) Counterfactual Reasoning.** This component is designed to learn a *causal model* $\vartheta$ for counterfactual prediction. It inputs a causal graph $G$, the dataset $D_{KG}(C, P_C)$, and produces the counterfactual outcomes of patients (units). As a proof of concept, we use CBN as a *causal model* $\vartheta$. Let $G = (X, E')$ be the *causal graph*, where $X = P_C$, and $(G, P_t)$ be the CBN based on $G$ trained over the dataset $D_{KG}(C, X)$. For a patient $e \in V$, whose treatment is $(e, p_T, t)$ and context is $c = \phi(e, P_C(p_T, p_Y))$. The counterfactual $p_Y$ of $e$ under a hypothetical treatment $(e, p_T, t') \notin E$ is $(e, p_Y, y')$ s.t. $P_{t'}(y'|x) \geq max(\{P_{t'}(y|x)\}_{y \in V_Y})$, where $V_Y = \{y | (\exists e' \in V)[(e', p_Y, y) \in E]\}$ that are all unique values of $p_Y$, and $P_t(y|c) = \frac{\sum_s P_t(y,c,s)}{\sum_{y,s} P_t(y,s)}$ is an interventional distribution derived from the CBN.

## 4 EXPERIMENTAL STUDY

In this study, we investigate the impact of smoking habits (treatment) on the biomarker results (outcome) of NSCLC patients. We evaluate the performance of HealthCareAI in the tasks of causal discovery and counterfactual prediction over synthetic KGs.

**Research Questions**. Let $KG$ be a KG of NSCLC patients. We raise two research questions. **Q1**: Given a causal graph $G^*$ provided by an expert (a.k.a. the **Expert** DAG in Figure 3), which encodes the causal relationships between properties of patients. Can our method discover the same DAG? **Q2**: Given counterfactual biomarkers of patients in $KG$, is our method able to correctly predict counterfactuals using the context of patients in $KG$?

**Original NSCLC KG**. The original NSCLC KG is reported in [1, 27]. In this study, we employed a fragment of the KG, containing 1,808 patients of class LCPatient. Each patient has the following properties: Biomarker that is either *ALK or EGFR* and *other biomarkers*, Age that is categorized as *Young (≤ 50 years)* or *Old (> 50 years)*, Gender that is *Male* or *Female*, SmokerType that is *Non-Smoker* or *Smoker*, FamilyCancer which is *OnlyMajor* if a patient's family antecedents have only these cancers: *Breast, Lung, Colorectal, Head*

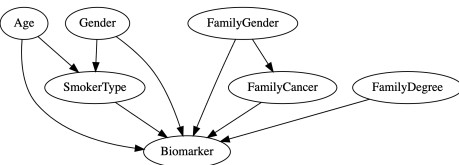

**Figure 3: Expert Designed Causal Graph (DAG) $G^*$ built over the properties of NSCLC patients in synthetic KGs**

*and neck*, *Uterus/cervical*, *Esophagogastric*, *Prostate*, otherwise *has-Minor*, FamilyGender that is *Women* if all family antecedents are women, *Men* if all of them are men, otherwise *WomenorMen*, and FamilyDegree that indicates the degree of relationship of family antecedents to the patient (i.e., first, second, or third degree).

**Synthetic NSCLC KGs**. We use an ANM [14], trained based on the **Expert** DAG $G^*$ (in Figure 3), using the dataset derived from the KG fragment (including 1808 patients) to faithfully capture the causal mechanism in the dataset. Using this model, we generate synthetic KGs with various patient numbers $N \in \{2k, 5k, 10k\}$. Specifically, Age, Gender, FamilyGender, and FamilyDegree are simulated from uniform distributions; while SmokerType, FamilyCancer, and Biomarker are simulated using logistic functions:

$$Y = 1/(1 + exp(-(\alpha + \beta \cdot X' + N(0, \sigma^2)))) \qquad (1)$$

where $Y$ is the synthetic variable; $X'$ represents parent variables of $Y$ in the DAG $G^*$ (see Figure 3). The $\alpha$ and $\beta$ are learned from the dataset using a logistic regression. A noise term $\mathcal{N}(0, 0.1^2)$ is applied to simulate other unobserved factors. Additionally, each synthetic KG includes the metadata of the original KG.

**Counterfactual Simulation**. Let $Y$=1 denote that Biomarker ($p_Y$) is *ALK or EGFR*, $T$=$t$ signify the treatments on SmokeType ($p_T$): *Non-Smoker* ($t$=1) and *Smoker* ($t$=0). Given a synthetic KG $KG = (V, L, E)$, a patient $e \in V$, whose treatment is $(e, p_T, t) \in E$. Let $S$ be a set $\phi(e, P_C(p_T, p_Y)) \cup \{(e, p_T, t')\}$ s.t. $(e, p_T, t') \notin E$, the counterfactual Biomarker of $e$ is generated using the Function (1) of Biomarker with input of $X' = x'$ following the assignment $X'_1, \ldots, X'_k := x'_1, \ldots, x'_k$ s.t. $(\forall i \in [1, k])[(e, X'_i, x'_i) \in S]$ where $k$ is the variable number of $X'$.

**Metrics**. For **Q1**, given the **Expert** DAG $G^*$ and an estimated DAG $G$, we use the *Jaccard Index* [16] to evaluate the performance: $JI(G, G^*) = \frac{|edges(G) \cap edges(G^*)|}{|edges(G) \cup edges(G^*)|}$. For **Q2**, given the ground truth counterfactual Biomarker of all patients $Y^{CF}$, we evaluate the performance of each CBN using the *Pearson correlation coefficient* [9] (PCC): $PCC(\hat{Y}^{CF}, Y^{CF}) = \frac{cov(\hat{Y}^{CF}, Y^{CF})}{\sigma_{\hat{Y}^{CF}} \sigma_{Y^{CF}}}$, where $\hat{Y}^{CF}$ represents the counterfactuals estimated by a CBN, $cov(\cdot)$ and $\sigma$ represent covariance and standard deviation. Higher values of these criteria indicate better performance.

**Compared Methods**. We compare **HealthCareAI** with all data-driven methods: **PC** [25], **FCI** [25], and **GES** [8] (implemented with the Causal-learn Python package [30]) and metadata-driven method: **LLM** (via official website of GPT-4 [20]). **HealthCareAI** is implemented using **PC** and GPT-4. CBNs of all methods are implemented using the pgmpy Python package [3]. Appendix A shows the LLM prompt for query causal relationships. The code and data for reproduction are available here [1].

---

[1]https://figshare.com/s/c50b45f2483987664c1c

**Table 1: The performance (*Jarccard Index*) of different Methods in causal discovery, compared against the Expert DAG $G^*$, using the synthetic KGs with various patient numbers $N$.**

| Method | $N = 2k$ (%) | $N = 5k$ (%) | $N = 10k$ (%) |
|---|---|---|---|
| **PC** | 55.6 | 66.7 | 66.7 |
| **FCI** | 55.6 | 66.7 | 66.7 |
| **GES** | 33.3 | 55.6 | 55.6 |
| **LLM** (GPT-4) | 66.7 | 66.7 | 66.7 |
| **HealthCareAI** | **88.9** | **100.0** | **88.9** |

**Table 2: Performance of different methods in counterfactual reasoning using CBNs based on DAGs produced by them. The PCC metrics are presented as the mean (± standard deviation), computed over a 5-fold cross-validation partitioning all NSCLC patients and their counterfactuals.**

| Method | $N = 2k$ (%) | $N = 5k$ (%) | $N = 10k$ (%) |
|---|---|---|---|
| **PC** | **95.0 (±2.5)** | 95.9 (±08) | 95.1 (±0.8) |
| **FCI** | **95.0 (±2.5)** | 95.9 (±0.8) | 95.1 (±0.8) |
| **GES** | 89.9 (±1.6) | 95.9 (±0.8) | 95.1 (±0.8) |
| **LLM** (GPT-4) | 89.9 (±1.6) | 90.7 (±1.6) | 90.0 (±0.9) |
| **HealthCareAI** | 93.9 (±2.7) | **96.4 (±0.7)** | 95.1 (±0.8) |
| **Expert** | 81.4 (±2.9) | **96.4 (±0.7)** | **95.9 (±0.6)** |

**Results of Causal Discovery**. We evaluate the DAGS produced by all methods against the **Expert** DAG $G^*$ (in Figure 3) over various synthetic KGs with different patient numbers $N$. Table 1 presents the evaluation results (by *Jaccard Index*) of different methods. The results show that the **HealthCareAI** outperforms other methods with *Jarccard Index* of 88.9% ($N$=2k), 100% ($N$=5k), and 88.9% ($N$=10k), followed by **LLM** with *Jarccard Index* of 66.7% in all settings of $N$. The **PC** and **FCI** methods, perform better than the **GES** method. Surprisingly, when $N \geq 5k$, increasing $N$ does not further improve the performance of causal discovery. These results answer **Q1** that our method achieves excellent performance in causal discovery. The results also show that the metadata-driven model **LLM** can act as a complement to the data-driven model, highlighting the potential of combining data and metadata from KGs for causal discovery.

**Results of Counterfactual Prediction**. Based on the DAGs estimated in the previous step, we train CBNs from the dataset over the synthetic KGs using the *Maximum Likelihood Estimation*. We denote the estimated counterfactual Biomarker by CBN as $\hat{Y}^{CF}$. For each patient $i$, $\hat{Y}_i^{CF}$ is 1 (represents *ALK or EGFR*) if $P^{CF}(Y$=1$|T$=$t, X$=$x)$ > 0.5, otherwise is 0 (represents *other biomarkers*). The evaluation results are presented in Table 2; the PCC metrics for each CBN are presented as the mean (± standard deviation), using 5-fold cross-validation on all NSCLC patients and their counterfactuals. The results indicate that **PC** and **FCI** outperform others in scenarios with limited data (i.e., $N = 2k$). In contrast, the CBN trained based on the **Expert** DAG $G^*$ exhibits the lowest performance, which may be explained by the overfitting issue [12]. This is likely due to the complex structure of the **Expert** DAG, which requires a large dataset to learn the conditional probability tables of the CBN. Conversely, the simpler structures of the DAGs derived from the **PC** and **FCI** methods allow for effective learning with less data.

In the scenario of large datasets (i.e., $N \geq 5k$), the Expert CBN outperforms all CBNs trained based on DAGs by other methods. Although enlarging the dataset generally enhances the generalization capability of CBNs, it is notable that the performance of all CBNs slightly declines as $N$ increases from 5k to 10k. The performance of the **LLM** CBN (using DAG by **LLM**), despite its high structural similarity to the **Expert** DAG, remains the worst across all datasets. This result underscores the limitations of DAGs estimated without considering the underlying data, emphasizing the crucial role of data-driven causal graph estimation for robust counterfactual reasoning By incorporating the causal relationships estimated by the constraint-based method **PC** and those inferred by **LLM**, the CBN produced by our **HealthCareAI** method achieves competitive performance across all settings of $N$, compared to the CBN based on the **Expert** DAG. The results address **Q2**, confirming that our method can provide good counterfactual reasoning performance.

## 5 RELATED WORK

**Causal Discovery**. Understanding causal relationships in data is crucial in healthcare [24]. Traditional approaches like the **PC** (Peter-Clark algorithm) and **FCI** (Fast Causal Inference) [25] rely on statistical tests to identify conditional independence and create the directed acyclic graph (DAG); the **GES** (Greedy Equivalence Search) [8] uses a score-based strategy to optimize the Bayesian Information Criterion (BIC) [8] to find the best DAG. Recent advanced methods have tried to use large language models for causal discovery [28].
**Causal Models**. Causal Bayesian Networks (CBNs) [21] and Structural Causal Models (SCMs) [21] are well-known causal models, where CBNs focus on causality within categorical data, enabling intervention analysis and interpretable inference; SCMs provide a general framework with structural equations for causal reasoning. Both models use DAGs to depict causal relationships, but CBNs are more suitable for categorical variables and easier to interpret.
**AI for healthcare**. Recent advanced AI techniques [5, 6, 10] in healthcare apply various machined learning or deep learning models for prediction, offering great potential in healthcare, suggesting that AI can perform as well as or better than humans in some tasks, e.g., diagnosing disease. However, the potential of causality [22] is largely overlooked. To bridge the gap, we propose a hybrid method for causal discovery and reasoning over healthcare KGs.

## 6 CONCLUSION AND FUTURE WORK

In this paper, we propose hybrid AI methods, **HealthCareAI**, which incorporate advanced ML techniques with domain knowledge for causal relationship discovery and offer counterfactual reasoning over KGs. Through extensive experiments on synthetic KGs of non-small cell lung cancer (NSCLC) patients, we demonstrate the effectiveness of **HealthCareAI** in discovering causal relationships and predicting counterfactual outcomes. These capabilities underscore the potential of **HealthCareAI** to enhance clinical decision-making and deepen the understanding of personal clinical results. Currently, **HealthCareAI** is tailored to work with categorical data, which, while useful, limits its applicability across broader data types, such as numerical data. We plan to extend its capabilities to analysis using multiple data types and reasoning processes from biomedical ontologies. Additionally, we want to explore various prompting engineering techniques for improving causal discovery with large language models (LLMs).

## ACKNOWLEDGMENTS

This work is partially funded by the project TrustKG-Transforming Data in Trustable Insights with grant P99/2020.

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

## A  GPT-4 PROMPT FOR CAUSAL DISCOVERY

```
# ROLE #
Act as an expert in lung cancer research,
    focusing on identifying causal
    relationships between properties within a
     knowledge graph of non-small cell lung
    cancer (NSCLC) patients.

# CONTEXT #
You are examining the causal relationships
    among properties of `NLCPatient` class,
    within a knowledge graph,
with rich metadata that describe their human-
    understandable label (`rdfs:label`),
    human-understandable meaning (`rdfs:
    comment`), domain (`rdfs:domain`), and
    range (`rdfs:range`):

1. Property: `Biomarker`
   - `rdfs:label`: "biomarker test result"
   - `rdfs:comment`: "The biomarker test
       results of NSCLC patients, including
       ALK or EGFR; 'other biomarker'
       includes MET, HER2, FGFR1, KRAS, RET,
       PDL1, HER2Mut, ROS1, BRAF."
   - `rdfs:domain`: `NLCPatient`
   - `rdfs:range`: `xsd:string`

2. Property: `Gender`
   - `rdfs:label`: "gender"
   - `rdfs:comment`: "The gender of NSCLC
       patients, either male or female."
   - `rdfs:domain`: `NLCPatient`
   - `rdfs:range`: `xsd:string`

3. Property: `SmokerType`
   - `rdfs:label`: "smoking habits"
   - `rdfs:comment`: "The smoking habits of
       NSCLC patients, classified as 'Non-
       Smoker' or 'Smoker'."
   - `rdfs:domain`: `NLCPatient`
   - `rdfs:range`: `xsd:string`

4. Property: `Age`
   - `rdfs:label`: "age"
   - `rdfs:comment`: "The age of NSCLC
       patients, classified as 'Young' (<= 50
        years) or 'Old' (> 50 years)."
   - `rdfs:domain`: `NLCPatient`
   - `rdfs:range`: `xsd:integer`

5. Property: `FamilyCancer`
   - `rdfs:label`: "family cancer type"
   - `rdfs:comment`: "The type of cancer in
       the family of NSCLC patients, either '
       OnlyMajor' which represents cancer
       types in {'Breast', 'Lung', '
       Colorectal', 'Head and neck', 'Uterus/
       cervical', 'Esophagogastric', '
       Prostate'} or 'hasMinor' which
       represents other cancer types."
```

```
   - `rdfs:domain`: `NLCPatient`
   - `rdfs:range`: `xsd:string`

6. Property: `FamilyGender`
   - `rdfs:label`: "family gender"
   - `rdfs:comment`: "The gender of NSCLC
       patients' cancered family antecedents,
        either 'Women', 'Men', or 'WomenorMen
       '."
   - `rdfs:domain`: `NLCPatient`
   - `rdfs:range`: `xsd:string`

7. Property: `FamilyDegree`
   - `rdfs:label`: "family degree"
   - `rdfs:comment`: "The family degree of
       NSCLC patients' cancered familial
       antecedents, classified as 'First
       degree', 'Second degree', or 'Third
       degree'."
   - `rdfs:domain`: `NLCPatient`
   - `rdfs:range`: `xsd:string`

# OBJECTIVES #
1. Analyze all these properties in the #
   CONTEXT # to identify all possible causal
    relationships among them.
2. Each identified causal relationship should
   be supported by evidence from academic
   studies, referenced using title, DOI or
   PMCID to ensure confidentiality and
   verify the reliability of the studies.

# INSTRUCTIONS #
1. Examine and understand the metadata of each
    property carefully.
2. Identify all possible causal relationships
   among the properties in the # CONTEXT #,
   supported by the title, DOI or PMCID of
   relevant academic research.
3. Provide, in a code box, the identified
   causal relationships in the previous step
    as a Python set of tuples, each with
   format: ([A], [B]) representing a causal
   relationship from property [A] to
   property [B].
```

**Listing 1: GPT-4 Prompt used in Our Case Study.**