# OpenReview forum: "Hybrid AI Approach for Counterfactual Prediction over Knowledge Graphs for Personal Healthcare"
_KDD.org/2024/Workshop/AIDSH — KDD-AIDSH 2024 Oral_

### Official Review · Reviewer_DhvC · 2024-06-17
**Hybrid AI Approach for Counterfactual Prediction over Knowledge Graphs for Personal Healthcare**

**Rating:** 6
**Confidence:** 4

**Review:**

**Summary**
The paper designs a framework combining symbolic reasoning over knowledge graphs (KGs), large language models (LLMs), and causal reasoning techniques to infer causal relationships and predict counterfactual. The proposed hybrid method achieves best performance in causal discovery task.

**Strength and weakness**
**Strengths**
- The proposed method has strong ability for causal discovery.
- Assist the conventional causal discovery methods with LLM to enhance the performance.

**Weakness**
- The evaluations primarily use synthetic datasets. While this is necessary for privacy reasons, the realism and complexity of real patient data can be substantially different. This could limit the applicability of the proposed methods to actual healthcare settings where data distributions and patient characteristics are more variable.
- While the paper aims at improving interpretability of AI in healthcare, the actual usability and interpretability of the outputs for clinical decision-making are not thoroughly assessed.
- There are some typos. For instance, in line 280, KG repeats twice.

---

### Official Review · Reviewer_kLjM · 2024-06-18
**Simple and effective method, Open-sourced code and data for easy reproduction**

**Rating:** 8
**Confidence:** 2

**Review:**

Summary Of Strengths:
- Clear motivating example and well-defined problem
- Simple and effective hybrid method for counterfactual prediction on personal healthcare
- Open-sourced code and data for easy reproduction of the result

Summary Of Weaknesses:
- Minor types need to be fixed, for example, Table 2 is too wide
- Can this method be effectively extended for other healthcare conditions and applied to real-life data

---

### Decision · Program_Chairs · 2024-06-28

Accept (Oral)